# Toward Learning Distributions of Distributions

Moritz Wohlstein[*1] and Ulf Brefeld[1]

[1]Leuphana Universität Lüneburg
 {moritz.wohlstein, ulf.brefeld}@leuphana.com

## Abstract

We propose a novel generative deep learning architecture based on generative moment matching networks. The objective of our model is to learn a distribution over distributions and generate new sample distributions following the (possibly complex) distribution of training data. We derive a custom loss function for our model based on the maximum mean discrepancy test. Our model is evaluated on different datasets where we investigate the influence of hyperparameters on performance.

## 1 Introduction

Neural networks have been shown to be universal function approximators in the limit of infinite layer width [1] or depth [2]. Particularly generative neural networks make use of the full power of neural networks by transforming a simple prior distribution, which allows for efficient sampling, into an arbitrarily complex data generation distribution. Within the domain of deep generational models, different concepts were developed over the years such as variational autoencoders (VAE) Kingma and Welling [3]. They belong to the family of variational Bayesian methods and can generate new instances by sampling from a distribution over the latent space of the autoencoder by making use of the reparameterization trick. Alternatively, flow-based generative models capture complex distributions by transforming simpler ones using bijections and leveraging the Change of Variables Theorem [4]. Especially in the context of image generation, diffusion models [5] have become prominent. Borrowing concepts from the area of non-equilibrium thermodynamics, they learn to denoise images so that, beginning with noise-only, the image is iteratively denoised until a clear image is obtained.

In this paper, however, we focus on Generative Adversarial Networks (GANs) [6]. These networks consist of a *generator* and a *discriminator* network. The *generator* uses the beforementioned statistical transformations to generate new data. The *discriminator* then acts as a classifier and aims to separate original training from generated data. The networks are trained together in a zero-sum game, in which the generator tries to "fool" the discriminator into incorrectly classifying generated data points as true ones, and the discriminator tries to correctly classify generated and true samples.

An interesting model related to GANs is the *generative moment matching network* (GMMN) [7, 8]. While the generator remains similar or even unchanged to that of regular GANs, the discriminator is replaced by the *maximum mean discrepancy* (MMD) [9] to measure distances between distributions $p$ and $q$, given by

$$\text{MMD}[\mathcal{F}, p, q] := \sup_{f \in \mathcal{F}} (\mathbf{E}_{x \sim p}[f(x)] - \mathbf{E}_{y \sim q}[f(y)]).$$

The MMD, unlike other methods, such as the Kullback-Leibler divergence, does not rely on density estimates of the distributions, but (implicitly) embeds them into a reproducing kernel Hilbert space in which the distance is determined. This renders the MMD particularly useful in settings dealing with sets of (generated) instances.

When the function space $\mathcal{H}$ is a *universal* RKHS [10] on a compact domain $\mathcal{X}$ of distributions $p, q$, the MMD fulfills all the properties of a metric [9], in particular the correspondence

$$\text{MMD}[\mathcal{F}, p, q] = 0 \Leftrightarrow p = q. \tag{1}$$

However, to devise a generative model that samples distributions based on a custom MMD loss while preserving the convergence guarantees as portrayed in Dziugaite et al. [7] and Li et al. [8], we need to devise a universal kernel that acts on the set of probability measures.

In this paper, we develop a new model based on the GMMN architecture, capable of generating not only distributions of vectors, but *distributions of distributions*. To achieve this goal, we devise an MMD between distributions of distributions and develop a new generator architecture based on *Hypernetworks* [11]. The proposed architecture allows for a fully implicit learning procedure, where the structures of the distributions are parameterized by neural networks, and the loss function is able to compare arbitrary statistical moments of the set of distributions.

The remainder is structured as follows. We present our main contribution, the derivation of our GMMN for distributional data, in Section 2 and report on empirical results in Section 3. Section 4 summarizes related work and Section 5 concludes.

---

*Corresponding Author.

Proceedings of the 6th Northern Lights Deep Learning Conference (NLDL), PMLR 265, 2025.

## 2 Main Contribution

### 2.1 Preliminaries

We are concerned with the problem of learning to generate new samples from a distribution over a set of datasets. Given a sample set of datasets, our goal is to construct a model that is able to approximate the underlying sampling distribution by minimizing a discrepancy between the distribution parameterized by the model and the distribution underlying the data. To achieve this, we apply the concept of maximum mean discrepancy to our setting. Since the entities of interest are empirical distributions represented by different datasets, we use universal kernels [10] on the set of probability measures.

Christmann and Steinwart [12] introduce universal kernels on non standard input spaces. They show that for a compact metric space $\mathcal{X}$ and a separable Hilbert space $\mathcal{H}$ with an injective map $\rho : \mathcal{X} \to \mathcal{H}$, one can derive a universal Gaussian-type RBF-kernel $k_\sigma : \mathcal{X} \times \mathcal{X} \to \mathbb{R}$ with

$$(x, x') \mapsto \exp(-\sigma^2 \|\rho(x) - \rho(x')\|_{\mathcal{H}}^2), \qquad \sigma > 0.$$

The authors further derive a kernel between probability distributions, based on the Fourier transform

$$k_\sigma(P, P') := \exp(-\sigma^2 \|\hat{P} - \hat{P}'\|_{L_2(\mu)}^2) \qquad (2)$$

Where the distributions $\hat{P}$ and $\hat{P}'$ are Fourier transforms of elements of the set $\mathcal{X} := \mathcal{M}_1(\Omega)$ of probability measures on a compact set $\Omega \subset \mathbb{R}^d$, and $\mu$ is a finite Borel measure on $\mathbb{R}^d$ with support$(\mu) = \mathbb{R}^d$.

Based on this similarity measure between distributions, we will derive an MMD between distributions of distributions, which is then used as the loss function of a generative model.

### 2.2 Motivation

To solve the problem described above, we define a training objective in form of an MMD loss $\mathcal{L}(y, \hat{y})$, a function of the generated samples and the training data. Our approach to generate samples is to transform a prior distribution from which we can easily sample new data points into the distribution over sample sets $q$ by using a deep neural network as the transformation function $G_\theta$. Our generation process consists of two stages. First, we need to sample a representation of the distribution $Q_l$ for each sample set we want to generate. We do this by modeling each $Q_l$ by a neural network $M_{w_k}$. The weight vectors $w_k$ of the network are sampled i.i.d. from a distribution that is parameterized by a second neural network $H_\theta$. In the second step, we sample datasets from each $Q_l$ independently, by transforming samples from a prior distribution $q^*$ using the

function $M_{w_k}$. This process produces a dataset $\hat{y}$, comprising a set of tuples of vectors $\hat{y}_i^j$

$$\hat{y} = \left( \left( \hat{y}_1^1, \hat{y}_1^2, \ldots, \hat{y}_1^{s_1} \right), \ldots, \left( \hat{y}_m^1, \hat{y}_m^2, \ldots, \hat{y}_m^{s_m} \right) \right)$$

where $\hat{y}_k^i = M_{w_k}(x_k^i)$, $w_k \sim H_\theta$ , $x_k^i \sim q^*$ and $s_k, m \in \mathbb{N}^+$. To train our model, we first randomly sample a new dataset $\hat{y} \sim q$ that we compare with the training data $y$ using the loss function $\mathcal{L}$. By choosing a differentiable loss function, we are able to use backpropagation and gradient descent algorithms to update the weights $\theta$ of our network. Next, we will derive a suitable differentiable loss function based on the concept of the maximum mean discrepancy test.

### 2.3 Distribution MMD

We can derive an empirical estimate of a universal kernel on empirical distributions of probability measures using the kernel defined in Equation (2). Given a training set $Y = (y_1, y_2, \ldots, y_n), n \in \mathbb{N}$, sampled i.i.d. from a distribution $P$, an empirical approximation $\tilde{P}$ of the distribution can be calculated as the sum of Dirac delta distributions centered on the elements of $Y$. Applying the Fourier transformation yields its characteristic function $\hat{\tilde{P}}$:

$$\tilde{P}(z) = \frac{1}{n} \sum_{i=1}^{n} \delta(y_i - z) \qquad (3)$$

$$\hat{\tilde{P}}(t) = \int e^{i\langle z,t \rangle} \tilde{P}(z) dz = \frac{1}{n} \sum_{i=1}^{n} e^{i\langle y_i, t \rangle}. \qquad (4)$$

The probability density of the zero-centered multivariate Gaussian distribution with the diagonal identity covariance matrix $\Sigma = \lambda \mathbf{1}$ is given by

$$f(x) = (2\pi\lambda)^{-d/2} \exp\left(-\frac{\|x\|_2^2}{(2\lambda)}\right). \qquad (5)$$

Choosing this distribution as the Borel measure $\mu$ for our $L_2$ function space in Equation (2), we get the following approximation of our kernel $k_\sigma(P, P')$ based on the empirical distributions $\tilde{P}$ and $\tilde{P}'$, where $\tilde{P}'$ is defined analogously to $\tilde{P}$ as $\tilde{P}'(z) = \frac{1}{m} \sum_{k=1}^{m} \delta(y_k' - z)$:

$$k_\sigma(P, P') = \exp\left(-\sigma^2 \|\hat{P} - \hat{P}'\|_{L_2(\mu)}^2\right) \approx$$

$$\exp\left(-\frac{\sigma^2}{\lambda^{d/2}} \left( \frac{1}{n^2} \sum_{i,j=1}^{n} \int_{\mathbb{R}^d} e^{i\langle y_i - y_j, t \rangle} e^{-\frac{\langle t,t \rangle}{2\lambda}} dt^d \right.\right.$$

$$-\frac{1}{nm} \sum_{i=1}^{n} \sum_{l=1}^{m} \int_{\mathbb{R}^d} e^{i\langle y_i - y_l', t \rangle} e^{-\frac{\langle t,t \rangle}{2\lambda}} dt^d$$

$$-\frac{1}{nm} \sum_{i=1}^{n} \sum_{l=1}^{m} \int_{\mathbb{R}^d} e^{i\langle y_l' - y_i, t \rangle} e^{-\frac{\langle t,t \rangle}{2\lambda}} dt^d$$

$$\left.\left.+\frac{1}{m^2} \sum_{k,l=1}^{m} \int_{\mathbb{R}^d} e^{i\langle y_k' - y_l', t \rangle} e^{-\frac{\langle t,t \rangle}{2\lambda}} dt^d \right)\right) \qquad (6)$$

Each of the four resulting integrals can be solved using the Fourier transformation of multivariate Gaussians:

$$\frac{1}{\sqrt{2\pi}^d} \int_{\mathbb{R}^d} e^{-\frac{at^2}{2}} e^{-i\omega t} dt^d = \frac{1}{\sqrt{a}^d} e^{-\frac{\omega^2}{2a}}$$

By setting $a = \frac{1}{\lambda}$ and $\omega = y_j - y_i$ for the first term, adapting the other terms correspondingly, we obtain

$$k_\sigma(P, P') = \exp\left(-\sigma^2 \|\hat{P} - \hat{P}'\|^2_{L_2(\mu)}\right)$$

$$\approx \exp\left(-\frac{\sigma^2}{(2\pi\lambda)^{d/2}}\left(\frac{1}{n^2}\sum_{i,j=1}^n \sqrt{\lambda}^{d/2} e^{-\frac{\lambda(y_j - y_i)^2}{2}}\right.\right.$$

$$-\frac{1}{nm}\sum_{i=1}^n\sum_{l=1}^m \sqrt{\lambda}^{d/2}\left(e^{-\frac{\lambda(y_l' - y_i)^2}{2}} + e^{-\frac{\lambda(y_i - y_l')^2}{2}}\right)$$

$$\left.\left.+\frac{1}{m^2}\sum_{k,l=1}^m \sqrt{\lambda}^{d/2} e^{-\frac{\lambda(y_l' - y_k')^2}{2}}\right)\right) \tag{7}$$

$$= \exp\left(-\sigma^2\left(\frac{1}{n^2}\sum_{i,j=1}^n e^{-\frac{\lambda(y_j - y_i)^2}{2}}\right.\right.$$

$$-\frac{1}{nm}\sum_{i=1}^n\sum_{l=1}^m \left(e^{-\frac{\lambda(y_l' - y_i)^2}{2}} + e^{-\frac{\lambda(y_i - y_l')^2}{2}}\right)$$

$$\left.\left.+\frac{1}{m^2}\sum_{k,l=1}^m e^{-\frac{\lambda(y_l' - y_k')^2}{2}}\right)\right) \tag{8}$$

$$= \exp\left(-\sigma^2\left(\frac{1}{n^2}\sum_{i,j=1}^n e^{-\frac{\lambda}{2}\|y_i - y_j\|_2^2}\right.\right.$$

$$-\frac{2}{nm}\sum_{i=1}^n\sum_{l=1}^m e^{-\frac{\lambda}{2}\|y_i - y_l'\|_2^2}$$

$$\left.\left.+\frac{1}{m^2}\sum_{k,l=1}^m e^{-\frac{\lambda}{2}\|y_k' - y_l'\|_2^2}\right)\right), \tag{9}$$

a closed form solution for the empirical Fourier transformation based kernel $k_\sigma(P, P')$ between sample distributions given by

$$k_\sigma(P, P') \approx \exp\left(-\sigma^2\left(\frac{1}{n^2}\sum_{i,j=1}^n e^{-\frac{\lambda}{2}\|y_i - y_j\|_2^2}\right.\right.$$

$$-\frac{2}{nm}\sum_{i=1}^n\sum_{l=1}^m e^{-\frac{\lambda}{2}\|y_i - y_l'\|_2^2}$$

$$\left.\left.+\frac{1}{m^2}\sum_{k,l=1}^m e^{-\frac{\lambda}{2}\|y_k' - y_l'\|_2^2}\right)\right) \tag{10}$$

With $Q_i \sim q, P_i \sim p$ the MMD based loss function is now given as an empirical estimate of a squared MMD between empirical distributions over distributions:

$$\mathcal{L}_{MMD}(p, q) = \frac{1}{M^2 - M}\sum_{i \neq j}^M k_\sigma(P_i, P_j) \tag{11}$$

$$-\frac{2}{MN}\sum_{i,j=1}^{M,N} k_\sigma(P_i, Q_j) + \frac{1}{N^2 - N}\sum_{i \neq j}^N k_\sigma(Q_i, Q_j).$$

This is however a theoretical loss function, based on the assumption of having knowledge about the underlying distributions $P_i$ and $Q_j$, when in our experiments we only used empirical estimates of the distributions based on sets of data points sampled from the respective distributions. The approximation of the kernel function is given in Equation (10).

## 2.4 Generative Model

The generative model consists of two major components. The first network which generates the weights for the second network will be called Hypernetwork, in accordance with Ha et al. [11]. Accordingly, the second network is called the main network, since it performs the main task, generating samples from the different distributions, which are parameterized by its weights that were sampled by the Hypernetwork, see Figure 1.

By using the MMD as the loss function for our model, we build upon the ideas of the GMMN proposed by Li et al. [8] and Dziugaite et al. [7]. Instead of comparing sets of data points represented by vectors, we directly compare sets of datasets, using the custom distribution MMD defined in Equation (10).

We assume that the distributions underlying different datasets can have a shared structure that can be directly learned by the main network using layers with weights shared between all main networks. We call these parameters $\phi$, whereas the parameters generated by the Hypernetwork are denoted by $w_k$. The former weights can be directly trained using gradient descent and backpropagation. Algorithm 1 shows the general structure of the training process of the proposed generative network.

# 3 Empirical Results

## 3.1 Family of univariate Gaussians

Starting from a simple and easily interpretable case, our goal is to learn a distribution over the family of univariate Gaussian distributions using our proposed model. We generate a training dataset by sampling a set of sample distributions, where each of the datasets follow a normal distribution. The means $\mu$ and standard deviations $\sigma$ of the sample distributions are distributed independently in the following way:

$$\mu \sim \mathcal{N}(\mu_\mu, \sigma_\mu^2), \quad \log(\sigma) \sim \mathcal{N}(\mu_\sigma, \sigma_\sigma^2)$$

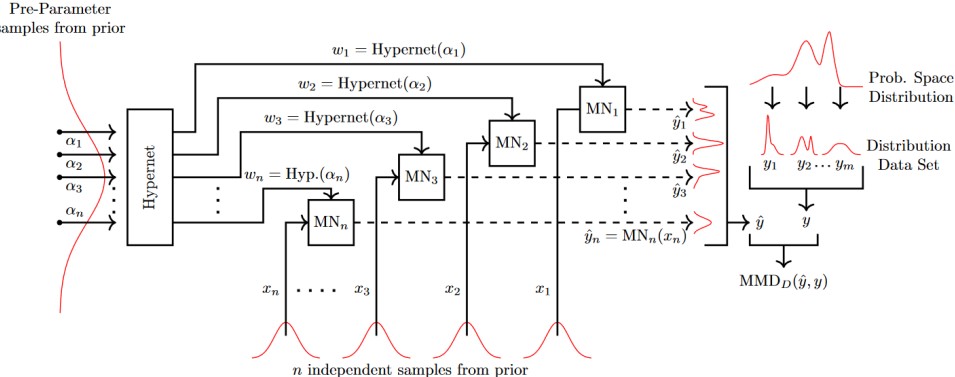

**Figure 1.** Architecture of the proposed generative model.

**Algorithm 1** Training the Hyper-GMMN

Parameters: $\theta, \phi$
Prior Distribution: $p^*$
Hypernetwork: $H_\theta$
Main Network: $M_{\phi, w_k}$
Data: $y = \left( \left( y_1^1, \ldots, y_1^{s_1} \right), \ldots, \left( y_n^1, \ldots, y_n^{s_n} \right) \right)$
**while** not converged **do**
  $m \leftarrow$ Determine $m$, $m \in \mathbb{N}^+$
  **for** $k \leftarrow 1$ to $m$ **do**
    $w_k \sim H_\theta$
    $m_k \leftarrow$ Determine $m_k(k)$, $m_k \in \mathbb{N}^+$
    **for** $s_k \leftarrow 1$ to $m_k$ **do**
      $x_k^{s_k} \sim q^*, \quad s_k \in \{1, 2, \ldots, m_k\}$
      $\hat{y}_k^{s_k} \leftarrow M_{\phi, w_k}(x_k^{s_k})$
    **end for**
  **end for**
  $L \leftarrow$ DistributionMMD$(\hat{y}, y)$
  $\theta \leftarrow \theta - \nabla_\theta L$
  $\phi \leftarrow \phi - \nabla_\phi L$
**end while**

We use our knowledge about the data generation process in the construction of our generative network, which acts as an inductive bias but significantly simplifies the network architecture to a one-layer main network with $n_{in} = n_{out} = 1$. We use a standard normal prior $\mathcal{N}(0, 1)$ to independently sample the seeds $\alpha_0, \alpha_1$. The Hypernetwork takes the form

$$H_{\mu_\mu, \sigma_\mu, \mu_\sigma, \sigma_\sigma}(\alpha) = \begin{pmatrix} \exp\left(\alpha_0 \sigma_\sigma + \mu_\sigma\right) \\ \alpha_1 \sigma_\mu + \mu_\mu \end{pmatrix} = \begin{pmatrix} \sigma \\ \mu \end{pmatrix}$$

and generates the "weight" $\sigma$ and the "bias" $\mu$ of the main network $M$

$$\hat{y} = M_{\mu, \sigma}(x_0) = x_0 \sigma + \mu$$

the main network generates the data by applying the reparameterization trick to data points $x_0$ sampled i.i.d. from a univariate standard normal distribution.

We generate data comprising 20 sample distributions of 20 samples each using the parameters

$\mu_\mu = 10, \sigma_\mu = 2.5, \mu_\sigma = 1, \sigma_\sigma = .5$. We use the MMD loss defined in Equation (11) and train the model using backpropagation and stochastic gradient descent for 10000 epochs. In each epoch, we generate a dataset of the same size as the training data. In Figure 2, the evolution of the Hypernetworks' parameters are plotted over the training process. These parameters correspond to the statistical moments of the parameters of the main network, and can be interpreted as the moments of the distribution over the sample distributions. The plots suggest a stable convergence to values close to the model parameters used in the data generation process. It can be seen that the bias w.r.t. the true parameters is smaller for lower order statistical moments.

## 3.2 Rotated Uniform

The second experiment is based on a dataset consisting of sets of data points uniformly distributed over a rectangular region of $\mathbb{R}^2$. The training data is generated by independently sampling the coordinates of vectors $z_i \in \mathbb{R}^2$ from one dimensional uniform distributions $z_i^0 \sim \mathcal{U}[0.2, 1]$, $z_i^1 \sim \mathcal{U}[-0.1, 0.1]$, and then multiplying each vector of the respective dataset by a dataset specific rotation matrix

$$A(\gamma) = \begin{pmatrix} \cos(\gamma) & -\sin(\gamma) \\ \sin(\gamma) & \cos(\gamma) \end{pmatrix}.$$

The dataset specific parameter $\gamma$, the angle of the rotation, is sampled from a uniform distribution over $[0, 2\pi]$. A set of 10 sample distributions generated is displayed in Figure 3. In our experiments, we use a training dataset set of 5000 sample distributions, comprising 50 data points each. Having equally sized sample sets is not required by our framework, but facilitates implementation.

To generate a set of datasets that matches the training set in distribution, we train a GMMN for distributions using the distribution MMD loss. The Hypernetwork has 4 hidden layers, with 20, 20, 50 and 100 nodes respectively. It samples the weights

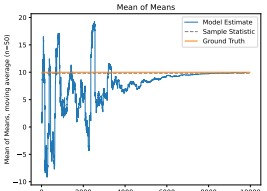
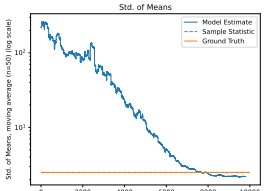
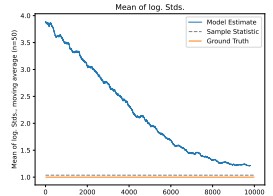
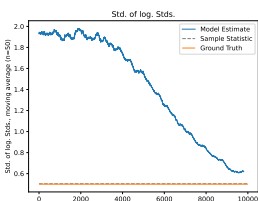

**Figure 2.** Evolution of Statistical moments

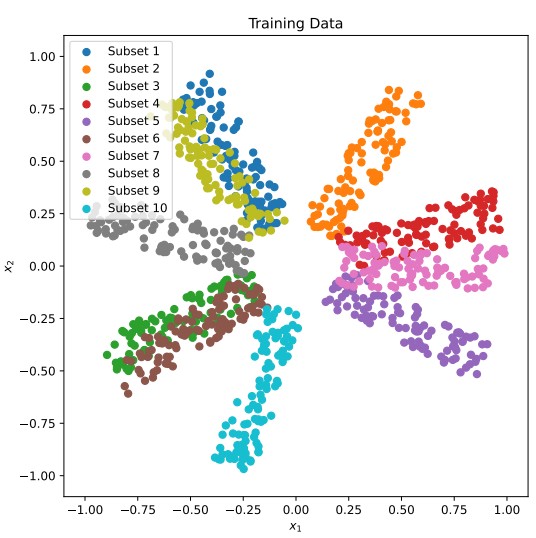

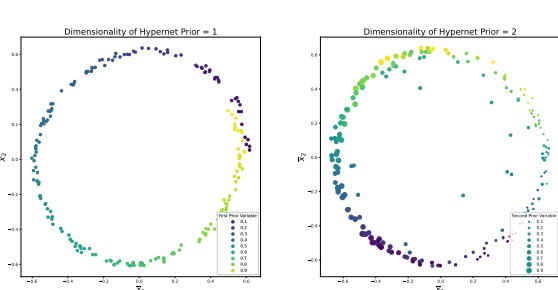

**Figure 3.** Visualization of the data

and biases of the second and third hidden layer of the main network with 10 and 5 nodes respectively, representing the individual distributions. The first layer of the main network comprises 10 nodes and shares its weights between all main networks; its weights are trained together with those of the Hypernetwork using the Adam optimizer introduced by Paszke et al. [13].

**MMD Hyperparameters** We are especially interested in the influence of the kernel width parameters of the MMD loss function on the quality of the generated distributions. We trained our model 10 times for different combinations of the distribution

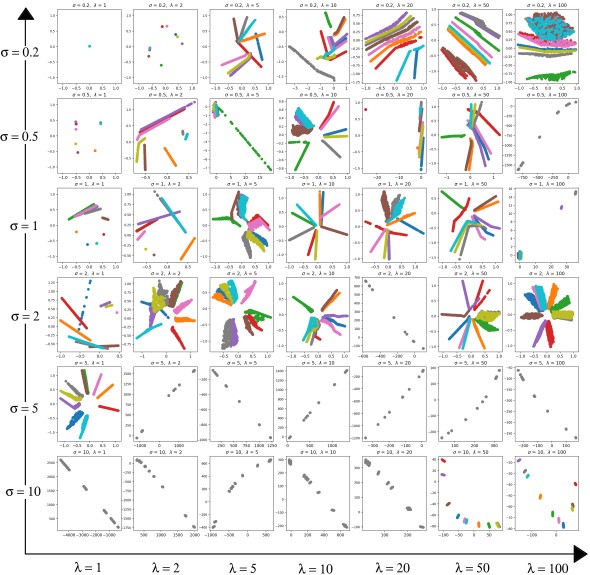

**Figure 4.** Results for $\sigma \in \{0.2, 0.5, 1, 2, 5, 10\}$ and $\lambda \in \{1, 2, 5, 10, 20, 50, 100\}$.

kernel width $\lambda$ and the parameter $\sigma$ of the kernel between data points. In Figure 4, generated data are plotted for each combination of the hyper parameters combination. For each setting, 10 random sample sets are displayed that were generated by the network achieving the minimal training loss in terms of distribution MMD.

One can see that our model has successfully learned to transform the prior distribution in a distribution over datasets that is very close to the training data. It is difficult to compare the performance of the algorithm for different hyperparameters $\sigma$ and $\lambda$, since they parameterize the loss function. Inspection by eye suggests optimal parameters of $1 \leq \lambda \leq 100$ and $0.5 \leq \sigma \leq 2$.

## 4   Related Work

There are multiple possible ways to approach the problem of modeling distributions of distributions and to generate samples from those models. One way is to use the concept of Dirichlet processes [14]. The realizations of this statistical process can be

seen as individual probability distributions. At every step in the sampling process, the next data point is sampled either from a continuous base distribution $H$ or, based on a weighting parameter $\alpha$, from one of the previously sampled values, based on their respective frequency.

Similarly, Gaussian processes allow the sampling of datasets following different distributions by sampling positions from a base distribution, which are then randomly but interdependently transformed using a randomly sampled Gaussian process.

The main advantage of our proposed method over the stochastic processes mentioned here is that the latter require explicit assumptions on the statistical model. Using the higher expressive power of neural networks, our neural network-based approach can approximate complex distributions effectively.

In general, conditional GANs [15] may offer an alternative to our approach. Conditional GANs are generative models that learn to generate distributions over a random variable $x$ dependent on an input $y$ by transforming a simple prior distribution into the desired (and possibly complex) conditional distribution $G(x|y)$. Our network also learns to parameterize distributions by generator functions, but, unlike conditional GANs, our network does not receive the dependency $y$ as input. Instead, our approach can be thought of as learning a probability distribution $P(y)$ jointly with $G(x|y)$.

Alternative approaches to learning distributions over distributions may be offered by other generative families, including variational autoencoders [3], flows [4], or diffusion models [5]. We settled mainly on GANs because the inclusion of a statistical test, which is key to adequately solving the problem, fits nicely into the framework as a surrogate discriminator.

## 5    Conclusions

We presented a promising proof of concept for learning distributions over distributions. Our approach relied on a Hypernetwork that parameterized a set of main networks that finally generated the desired data. All networks were trained simultaneously and the overall training loss is a tailored MMD-based loss function. Preliminary empirical results on small-scale toy data showed that the underlying concepts can indeed be learned, although stability of the training process is still an issue.

In this paper, our model was used to generate low-dimensional datasets leading to promising results.

Another limitation is the stability of the training process. Figure 4 clearly shows a strong correlation between the hyperparameter values and the outcome. While some settings lead to the desired distributions, others just produce nonsense. Although it is natural that hyperparameters of learning algorithms have to be adjusted to a problem at hand, a larger range of suitable parametrizations was preferable. Thus, evaluating different training algorithms or changing the distribution estimate from sums of Dirac distributions to kernel density estimates seem to be promising avenues to stabilize the training behavior for future work.

## Acknowledgments

This project was funded in parts by the German Federal Ministry of Education and Research (BMBF) under grant SynthiClick (16KISA110).

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
