# OpenReview forum: "Toward Learning Distributions of Distributions"
_NLDL.org/2025/Conference — NLDL 2025 Oral_

### Official Review · Reviewer_y1z6 · 2024-10-03
**Generating distributions using GANs and MMD**

**Confidence:** 3

**Summary:**

The authors tackle the problem of generating and sampling distributions (rather than the more vanilla problem of generating and sampling points from distributions). The generated distributions can then be used as generative models themselves. A GAN-like architecture is used with a discriminator and generator, where the discriminator loss is a squared estimate of an MMD. The MMD operates on distributions, and as such requires an appropriate notion of a kernel over the space of distributions.

There are a few relatively minor errors, and some odd descriptions of standard results (described below), however it is possible to see past these errors and broadly understand what the authors are doing. What is more challenging is understanding why the authors are tackling this particular problem. Only synthetic data is considered, and no real life use cases are tried or even discussed. My concern is that on any non-synthetic problem, the method would fail to perform well due to the difficult nature of the problem (in a sample complexity sense).

**Strengths:**

- The overall pipeline seems to be correct. Estimate a density using a sum of Dirac measures, compute its Fourier transform, and then pass this into an appropriate kernel over the space of probability measures. This kernel is used to estimate a squared MMD, which is used as the loss for the discriminator network.
- The method is mostly clear, with some isolated instances of inaccuracies (discussed below).
- To the best of my knowledge, the problem tackled is novel. I have not seen other works solve this problem, but this is also not my area of expertise.

**Weaknesses:**

- I am not sure what the first equality in equation (4) is saying (the second equality looks okay, though). Perhaps $\widetilde{P}(z)$ is missing from the integrand, or perhaps the measure $\mu$ is supposed to contain $\widetilde{P}$. Either way, it is not clear, as I can't easily see how the integrand depends on $y_i$.
- Equation (6) is unnecessarily long. As far as I understand, each of the integrals is just a Fourier transform. The text between (6) and line 211, first column, is even more confusing. A long description of the Fourier transform of a Gaussian measure is given, using what appears to be very complicated techniques, but the Fourier transform of a Gaussian measure is a very standard result (and is famously Gaussian itself, as the authors correctly derive). The whole pipeline is quite straightforward and the current set of equations hinder rather than help: Integrate the sum of Dirac masses against the complex exponential to obtain a sum of complex exponentials. Integrate these against a Gaussian measure to obtain a sum of Gaussian measures. These become arguments to a squared exponential kernel (which itself resembles a Gaussian measure).
- It is incorrectly stated that equation (10) is "the MMD". Equation (10) is a (biased) empirical estimate of a **squared** MMD between **empirical distributions**. The squared is important, but it is even more important to try and disambiguate the nature of "empirical distributions". Usually we estimate the MMD given samples from a distribution. But here you are first sampling an empirical distribution (which is itself an approximation), and then estimating the squared MMD on top of that using the usual technique. A few sentences would be helpful.
- There appears to be an index error in the second sum in equation (10) - $j$ is missing.
- Equation on line 258 appears to use nonstandard notation, which is confusing. Usually the second parameter of $\mathcal{N}$ is a variance, not a standard deviation.

- I don't see an open discussion of the limitations of the work, as described in the reviewer guidelines.
- I am not convinced about the experimental rigour. It appears as though for both experiments, one seed of random hyperparameters is tried for each hyperparameter setting.

Minor:
- wrong punctuation for "fool" on line 045. Should be ``fool''. Same for other quotation marks
- "MMD fulfils the properties of a metric [9], that is, (1)". This is correct, but potentially poorly worded. With a universal kernel, MMD fulfills all of the properties of a metric (there are more than one).
- Some unusual grammar line 158 - 160, second column. Maybe a full stop should be a comma?
- The imaginary unit $i$ clashes with some index notation. Consider using $j$ for such indices.
-  Figure 1 is too small to read without zooming in a lot. All figure texts and legends are actually too small.

Questions:
- Figure 2. Did you try and run the training process for longer to see if the last two parameters ever reach closer to their target?
- Why not use a better estimator for the empirical distribution? Dirac measures are known to be poor estimates - what about a kernel density estimator with a Gaussian measure? This should still admit a closed-form expression, because you will just be essentially still computing Fourier transforms of Gaussian densities.

**Justification:**

There are issues with correctness, which I would be willing to overlook and think the authors could address in a modified manuscript. What is more challenging is why the problem the authors consider is of interest. Right now only synthetic examples are considered, and it is not clear whether this approach would actually work in nonsynthetic data. My worry is that the sample complexity would render this approach intractable.

---

> ### Author Rebuttal · Authors · 2024-10-24
>
> # Equation 4
> Thank you for pointing out the unclarity in Equation 4. The measure $\mu$ stems from a previous version of this paper and is indeed the measure containing $\tilde{P}$. We will correct this in the revision of the paper. The integrand can be then seen as a sum of Fourier transforms of Dirac delta distributions, where the $y_i$ are the offsets of the delta Distributions.
>
> # Equation 6
> Thank you for pointing out that the derivation can be simplified by using the Fourier transformation of multivariate Gaussians, we will adjust our derivation in the revised version of the paper. We believe that Equation (6) remains nevertheless useful, since it shows why integrating the characteristic functions over the Borel measure is equivalent to performing a Fourier transformation of a sum of shifted versions of said measure.
>
> # Equation 10
> You are right that Equation (10) is not the MMD, we will be more careful with our formulation in the revised paper, and rightfully call it an ``empirical estimate of a squared MMD between empirical distributions over distributions". Further we will make clear, that the expression in Equation (10) is a theoretical loss function, that is based on the assumption of having knowledge about the underlying distributions $P_i$ and $Q_j$, when in our experiments we only used empirical estimates of the distributions based on sets of datapoints sampled from the respective distributions.
>
> # Index and Variance
> Good point, we will use the index $j$ in Equation (10) in the revised paper and will change $\mathcal{N}(\mu,\sigma)$ to $\mathcal{N}(\mu,\sigma^2)$.
>
> # Limitations
> At the moment, we put the limitations in the conclusion section to save on space, but we will highlight them in the revised version of the paper and include limitations in the generalization abilities of the current state of our method.
>
> # Hyperparameter Seed
> We are not sure what is meant by ``One seed of random hyperparameters is tried for each hyperparameter setting", since the performance of our results is shown for a set of hyperparameters in Figure (4), made up of all combinations of a broad, logarithmic range of values for $\lambda$ and $\sigma$.
>
> # Minor
> Thank you for the pointing out the errors, we will fix them in the revised version of the paper. We will think about making Figure (1) into a full-width image over both columns and putting larger versions of the figures into the appendix.
>
> # Questions
> Testing out different different estimators for the empirical distribution is a promising direction for future research, thank you for the suggestion.
>
> # Application to Real-World Data
> Consider a company with complex but private data that must not be shared. Third parties however would benefit from having access to that data. If the company could generate data that does not differ statistically from the original data, they could easily give the generated data away and everybody would be happy. One could think of associating the behavior of different people or other entities with one distribution each. Our approach is considered a first step towards generating arbitrarily complex data distributions that are not significantly different from original data.

---

### Official Review · Reviewer_yoAQ · 2024-10-09
**Sound approach to generative modeling of distributions**

**Confidence:** 3

**Summary:**

This paper considers the generative moment matching network (GMMN), the GAN framework with its discriminator defined by the maximum mean discrepancy (MMD). The authors propose the GMMN for generating distributions of distributions by developing the MMD between the distributions of distributions basically correctly.

**Strengths:**

The proposed method is sound and promising and its preliminary demonstration is illustrative.

**Weaknesses:**

The derivation might have a minor error on the discussion around eq.(8).

**Justification:**

The proposed approach is sound and is basically correctly derived.
- It might be nicer to explain an intuitive meaning of $\lambda$ in eq.(5), that is, what happens to the behavior of the proposed method when $\lambda$ is changed.
- Although the two terms in eq.(8) are summed into a single integral, I don’t think this is true while each of the two terms in fact goes to zero as $L\rightarrow \infty$.
- It might be nicer to discuss the difference as a methodology between the proposed method and a direct modeling of distributions of distributions like the Dirichlet process.

minor:
p.3, l.230: ``parameterized’’ is misspelled.
p.4, l.258: The mean of the Gaussian for $\log (\sigma)$ should be $\mu_{\sigma}$ (not $\sigma_{\mu}$).
p.4, l.296: The lower endpoint of the uniform distribution of $z_i^1$ may be $-0.1$.

---

> ### Author Rebuttal · Authors · 2024-10-24
>
> # Lambda Parameter
> Thank you for pointing us to the missing explanation of the lambda parameter in Equation (5), we will provide some context in the revision.
>
> # Equation 8
> Equation (8) is correct according to our understanding but actually written down somewhat sloppy. We will revise the presentation and hopefully clarify all questions by that.
>
> # Dirichlet Process
> We will add a comparison with DPs, thank you for pointing this out.

---

### Official Review · Reviewer_Xwtg · 2024-10-09
**Promising Architecture for Generating Distributions of Distributions, but Limited Experimental Validation and Contextualization**

**Confidence:** 3

**Summary:**

This paper proposes a novel generative model based on the generative moment matching network (GMMN) architecture, designed to generate distributions over distributions. The framework employs a Hypernetwork to generate model parameters, with each parameter set corresponding to a sampled distribution. The paper derives a loss function based on the Maximum Mean Discrepancy (MMD) test, enabling the networks to learn by comparing sets of datasets. The model’s effectiveness is illustrated through experiments on two small-scale toy datasets.

**Strengths:**

* The use of a Hypernetwork to generate parameters corresponding to distinct distributions is a straightforward approach. This design intuitively aligns with the task of learning a distribution over distributions.
* The visual representation of the model architecture is clear and effectively aids in understanding the proposed method, providing valuable insight into the model’s structure and the role of each component.

**Weaknesses:**

* The experiment section lacks robust evidence to convincingly demonstrate the proposed method’s performance:
    * The paper only includes experiments on two toy datasets, without any real-world datasets, which limits the generalizability and practical relevance of the results.
    * In Section 3.2, only one out of 42 tested hyperparameter pairs produces a distribution of distributions similar to the training set, calling into question the method’s robustness.
* The paper does not adequately situate the proposed method within the broader context of existing methods that model distributions over distributions. It is unclear where the method’s innovations lie, and the paper lacks a performance comparison with relevant baselines, such as the Dirichlet process.
* The paper does not sufficiently motivate the importance of modeling distributions over distributions. The transition in Lines 68-73 feels abrupt, and the lack of real-world examples or applications makes it difficult to appreciate the method’s broader relevance.
* Some minor issues:
    * Line 222, $(P_i, Q_j)$ instead of $(P_i, Q_i)$
    * Line 243: it’s unclear which network the parameter $\\phi$ belongs to, since we have $\\theta$ for the Hypernetwork, and $w_k$’s for the main networks.
    * Line 258: $\\mu_{\\sigma}$ for the mean of $\\log(\\sigma)$

**Final Rebuttal Confidence:**

3

**Final Rebuttal Justification:**

The paper appears incomplete in its current form:
* The study only includes toy examples as "proof of concept" demonstrations.
* There is limited exploration of the effects of hyperparameters on training stability and generalization to the test set, leaving the experimental results unconvincing in demonstrating the proposed method's effectiveness in learning distributions of distributions.
* The absence of baseline models limits the contextual understanding of the proposed method's performance.
* The lack of real-world applications weakens the motivation for the method and its potential impact.

Although the authors have responded to these concerns, I intend to maintain my original rating based on the paper's present state.

**Justification:**

Although the model architecture design effectively aligns with the goal of modeling distributions over distributions, the paper does not provide sufficient experimental evidence to convincingly demonstrate the efficacy of the proposed approach. The experiments are limited to two toy datasets, with no real-world examples, and the method’s robustness is questionable given that only one out of 42 hyperparameter pairs yielded results similar to the training set. Furthermore, the paper lacks contextualization within the existing landscape of methods for modeling distributions over distributions, offering no comparisons to established techniques like the Dirichlet process. Additionally, the paper does not adequately motivate the importance of the problem or provide real-life applications, which makes it difficult to assess the broader impact and innovation of the approach.

---

> ### Author Rebuttal · Authors · 2024-10-24
>
> # Performance
> At the moment, our model is a proof of concept, which we think shows a promising direction for future research. As such, we mainly focus on the theoretical background of the problem and chose to evaluate our method on easy to interpret datasets, which we can fully control.
> In this work, we mainly investigate the influence of the hyperparameters of the loss function. We are certain that a broader search over additional hyperparameters, including the network size and training parameters as batch size can show regions of higher training stability and generalization in the hyperparameter space. Investigating the influence of different parameters is an interesting topic for future research.
>
> # Context
> We will provide a comparison with the Dirichlet process in the revised paper, thank you for the suggestion. The Dirichlet process differs in some aspects from our concept. Our method does not need a base distribution to sample from, supports the generation of continuous distribution "out of the box" and can be easily trained using back propagation and gradient descent.
>
> # Motivation
> Consider a company with complex but private data that must not be shared. Third parties however would benefit from having access to that data. If the company could generate data that does not differ statistically from the original data, they could easily give the generated data away and everybody would be happy. Our approach is considered a first step towards generating arbitrarily complex data distributions that are not significantly different from original data.
>
> # Minor Issues
> We corrected the typos (thanks!) and will also adopt the proposed notation in the revision.
>
> We will make the difference between the $w_k$ and$\phi$ parameter clearer in the revision.

---

### Meta-Review · Area_Chair_kpGC · 2024-11-02

**Recommendation:** Accept (Poster)
**Confidence:** 5

**Metareview:**

This paper considers estimating distributions over data objects that are, themselves, distributions. This is an important problem that e.g. lies at the heart of information geometry, and has also been studied in various applications before, e.g. covering:
- Distributions of normal distributions, which have appeared in diffusion tensors in diffusion MRI (describing estimated probability distributions over white matter fiber orientations), covariance descriptors in computer vision (describing local feature distributions) and more
- Distributions of Gaussian processes describing e.g. uncertain estimates of white matter bundle trajectories or uncertain estimates of yearly temperature curves

In the last bullet, the interesting uncertainty is aleatoric, or irreducible -- and these examples are therefore special cases of the problem of incorporating irreducible data uncertainty into the downstream analysis. Distributions over the aleatoric distributions could be a useful tool in this regard.

The reviewers largely appreciate the paper as an interesting and clear proof of concept study, which is well suited to a conference. The most important highlighted concern being the relevance of the problem. The authors are encouraged to stress the motivation for the problem in their final study, as well as incorporate any concerns, questions and useful suggestions made by the reviewers.

**Suggested Changes To The Recommendation:**

2: I'm certain of the recommendation.  It should not be changed

---

### Decision · Program_Chairs · 2024-11-06

**Decision:**

Accept (Oral)

**Comment:**

We recommend an oral and a poster presentation given the AC and reviewers recommendations.